# The Impact of Jujube Witches’ Broom Phytoplasma on the Community Structure of Endophytes in Jujube

**DOI:** 10.3390/microorganisms13061371

**Published:** 2025-06-12

**Authors:** Nian Wang, Mengli Wang, Ziming Jiang, Wenzhe Zhang, Ziyang You, Xueru Zhao, Jia Yao, Chenrui Gong, Assunta Bertaccini, Jidong Li

**Affiliations:** 1College of Forestry, Henan Agricultural University, Zhengzhou 450046, China; 2Henan Academy of Forestry, Zhengzhou 450008, China; 3Department of Agriculture and Food Science, *Alma Mater Studiorum*-University of Bologna, 40127 Bologna, Italy

**Keywords:** biological control, high-throughput sequencing, microbial diversity, plant disease

## Abstract

Evidence from an increasing number of studies indicates that plant endophytic microorganisms play a significant role during biotic and abiotic stress resistance. To date, however, only a handful of studies on endophytes in response to the presence of phytoplasmas have been conducted. The production of jujube (*Ziziphus jujuba*) is threatened by jujube witches’ broom (JWB) disease, which is associated with the presence of the JWB phytoplasma ‘*Candidatus* Phytoplasma ziziphi’. To investigate the impact of jujube witches’ broom phytoplasma on the endophyte populations in jujube, high-throughput sequencing was performed in healthy and JWB-infected orchard jujube trees and *in vitro* jujube shoots. The results showed that the presence of JWB phytoplasma in jujube altered the abundance, diversity, and community structure of endophytic bacteria and fungi. In the branches and the roots, the presence of JWB phytoplasma was associated with an increase in the richness of the endophytic communities and a decrease in their diversity, with the phyla Proteobacteria, Firmicutes, and Bacteroidota and the genus ‘*Ca*. Phytoplasma’ becoming the most abundant. The presence of phytoplasmas was also associated with the remodeling of the endophytic microorganisms’ interaction network, shifting to a simpler biodiversity state. These results demonstrate the response of the jujube endophytic community to the presence of JWB phytoplasmas and shed light on the possible antagonistic agents that could be further evaluated for JWB disease biocontrol.

## 1. Introduction

Plant endophytes are microorganisms present in the organs of plants that do not cause direct harm to them; they are also present in fungal and bacterial asymptomatically infected host plants [1]. Plant endophyte interactions are crucial for nutrient acquisition, growth and development, and also enhance tolerance to various environmental stresses [2]. The presence of endophytes is influenced by numerous factors, including changes in abiotic and biotic factors such as temperature, humidity, soil fertility, and pathogenic microorganisms. Moreover, the growth and development stages of plants directly or indirectly induce changes in the community structure and function of endophytes in plants [3,4]. It has been shown in numerous studies focusing on interactions among plants, pathogens, and endophytes that some microorganisms may act as biological control agents for plant diseases. Through screening and competition selection of bacterial endophytes in healthy walnut trees, it was found that *Bacillus velezensis* strain JD81 and *Pseudomonas protegens* strain JD62 can be used to develop agents to manage the walnut bacterial canker [5]. Through 16S rRNA gene and ITS region sequencing, the bacteria *Sporolactobacillus* and *Stenotrophomonas* were found to accumulate in *Rhododendron delavayi* Franch roots and were beneficial against the root rot disease [6]. In another study, *Pseudomonas* isolates S02, S09, and S26 were selected with antagonistic activity against the patchouli wilt caused by *Ralstonia solanacearum* [7]. The beneficial and antagonistic screening of endophytes through high-throughput sequencing has drawn increasing attention to the biological control of plant diseases [8].

Phytoplasmas (Kingdom, Bacteria; Phylum, Firmicutes; Class, Mollicutes; Genus, ‘*Candidatus* Phytoplasma’) are Gram-positive bacteria without a cell wall [9]. Transmitted by insect vectors such as leafhoppers and planthoppers, phytoplasmas colonize the phloem of the host plant and the intestine, hemolymph, and salivary glands of the insect vectors [10,11]. Phytoplasmas are associated with over one thousand plant diseases worldwide. Compared to fungal and bacterial pathogens, evidence is lacking on the effects of these bacteria on the host microbiota. The presence of significant differences in the microbial community structures of symptomatic and asymptomatic leaf and branch samples of phytoplasma-infected Chinese chestnut yellow crinkle-diseased trees has been reported [9]. The findings of another study indicated that ‘*Ca*. P. mali’ can modify the composition of endophytic bacterial communities in infected apple trees [12].

Jujube (*Ziziphus jujuba*) is a perennial deciduous fruit tree species of the Rhamnaceae family. Originating in the middle and lower Yellow River basin in Northern China, jujube has a more than 8000-year history of cultivation and utilization [13]. It was introduced to countries neighboring China, namely, Japan and South Korea, over 2000 years ago. Jujube has now spread to more than 40 countries, including the United States of America, Australia, and European countries such as Romania, Italy, and Spain [14,15]. Jujube fruit can be consumed both fresh and dried and serves as a traditional herbal medicine in China. As a highly representative fruit tree species, jujube is recognized for its high economic value and ecological benefits. Jujube witches’ broom (JWB) is a deadly infectious disease associated with the presence of JWB phytoplasma (‘*Ca*. P. ziziphi’) [16,17]. JWB-phytoplasma-infected jujube trees exhibit altered fundamental plant development processes, decreased fruit quality, and ultimately, death within a few years of infection, resulting in severe losses to the jujube industry [18,19]. JWB phytoplasma is not available as a pure isolate on artificial media [14]. This factor has hindered the comprehension of its pathogenic mechanism and the development of appropriate control methods. Despite the fact that a number of studies have been conducted on the phytoplasma pathogenic mechanism and control methods, at present, there remains a lack of effective treatments, particularly in relation to biological control agents, for JWB disease.

To date, only a few studies on the interaction between pathogens and endophytes in jujube plants have been performed. Through the isolation of endophytes and nutritional competition screening with the jujube shrunken-fruit disease pathogen *Alternaria alternata*, a strain of *Bacillus subtilis*, St-zn-34, was found to exhibit strong inhibitory activity [20]. The application of the *Actinobacterium* strain Act12 to jujube through root drenching and foliar spraying significantly altered the diversity of soil microorganisms and enhanced the interaction networks among them. These changes led to increased soil nutrient availability and consequent improvement in longitudinal diameter, flower bud and fruit number, and individual fruit weight [21]. To date, however, no study has addressed the influence of JWB phytoplasmas on jujube endophyte diversity.

In this study, the high-throughput sequencing amplicon of the ITS and 16S rRNA gene was used to analyze the microbial community composition in different parts of both healthy and JWB-infected jujube trees. Simultaneously, the microbial community composition of healthy and JWB-infected *in vitro* jujube shoots was analyzed. The aim of this study was to evaluate the influence of JWB phytoplasmas on the composition, diversity, and potential function of the jujube endophytic community. The results of this study could be used to evaluate potential biological control agents for JWB disease. Moreover, some interaction relationships between the endophytic microbiome of jujube and the JWB phytoplasmas are also inferred.

## 2. Materials and Methods

### 2.1. Sample Collection

Samples were collected from eight-year-old jujube trees of the variety ‘Huizao’ in the experimental orchard in the suburbs of Zhengzhou city, together with *in vitro* shoots from the laboratory of Henan Agricultural University (Appendix A). The orchard was established on sandy soil, with an average annual rainfall of 670 mm and an average temperature of 14.7 °C. and it was equipped with a flood irrigation system. All of the trees were uniformly managed according to commonly employed practical methods [22]. The JWB disease occurrence rate in the orchard was 5%.

Six healthy and six JWB-diseased trees were randomly selected from the orchard in August 2023. The distance between the sampled trees was maintained at a minimum of 15 m (Appendix A). Leaves (HL and DL) and branches (HB and DB) of 1 cm diameter and roots (HR and DR) of 1 cm diameter in the 5–10 cm soil layer from different parts of the same tree were collected as one repeat. Three tissue-cultured healthy and three JWB-diseased shoots (TCHS and TCDS) were also employed. In total, the samples comprised six repeats of orchard trees and tree repeats for tissue-cultured shoots, for healthy and infected materials each. All the samples were verified as healthy (phytoplasma negative) or diseased (phytoplasma positive with JWB symptoms) through PCR with the phytoplasma universal primer pair U5/U3 (Appendix A) [23]. The samples were cleared from epiphytic microbes via sequential washing in sterile water for 30 s, 70% ethanol for 2 min, sodium hypochlorite solution (2.5% active Cl^−^ and 0.1% Tween 80) for 5 min, and 70% ethanol for 30 s, and rinsed 5 times in sterile water [24]. All samples were treated with liquid nitrogen immediately after surface sterilization and stored at −80 °C. For validation testing of surface sterilization, water collected from the final wash process was placed on a potato dextrose agar (PDA) medium and maintained at 30 °C for 48 h [25].

### 2.2. DNA Extraction and Sequencing

Total DNA was extracted from the jujube samples using a PowerSoil DNA Isolation Kit (MoBio, Banbury, UK, Catalog No. 12888-50). A two-step method was employed to construct the Metagenomic Cosmid DNA (MCD) library. During the first step, primers with adapters were designed for PCR using DNA as a template. During the second step, PCR was performed using the PCR products from the first step as templates [26]. PCR products were purified and recovered based on electrophoretic quantification (ImageJ software version 1.54f) results with a 1:1 mass ratio [27]. The constructed libraries were sequenced using Illumina NovaSeq 6000 PE250 (Illumina, San Diego, CA, USA) [28]. The MCD primer information is presented in Appendix A.

### 2.3. Sequence Analysis

Sequence analysis was performed using the Illumina NovaSeq sequencing platform [29]. Through paired-end sequencing, small-fragment libraries for sequencing were constructed. First, the raw reads were filtered using Trimmomatic (Version 0.33) software [30]; thereafter, the primer sequences were identified and removed using CUTADAPT software (Version 1.9.1) [31], and the clean reads without primer sequences were obtained using the DATA2 method in QIIME2 (Version 2020.6) [32]. Once the data were de-noised, 0.005% of all sequence numbers were filtered as the threshold amplicon sequence variant (ASV). The ASV abundance information was normalized using the sequence number standard corresponding to the least sequenced samples. Using SILVA (Release 138) [33] and UNITE (Release 8.0) [34] as a reference database and using a naive Bayes classifier to annotate feature sequences, an ASV annotated to multiple taxonomic units at a certain level was marked as unclassified. Taxonomic information for each feature was obtained; thereafter, the community composition of each sample was analyzed at different levels (phylum, class, order, family, genus, and species), and a species abundance table was generated using QIIME software. The α-diversity and β-diversity were analyzed based on normalized ASV abundance information.

### 2.4. Statistical Analyses

The α-diversity was analyzed by calculating the Chao1 and Shannon indices for sample species diversity, with differences in the α-diversity between distinct groups being analyzed using a Kruskal Wallis H test. To verify the differences in microbial communities between groups, the principal coordinate analysis (PCoA) of the Bray–Curtis distances at the OTU level was calculated [35]. In addition, the significance of the effect on β-diversity was measured using a PERMANOVA (Adonis, Tokyo, Japan). All these indices were calculated using the Vegan R package (Version 4.2.0) and visualized with the ‘Ggplot2’ R package [36]. The Venn diagrams were generated with the VennDiagram R package. Functional annotation of the endophytic bacterial and fungal community was performed with PICRUSt2 (Version 2.3.0) and FUNGuild software (Version 1.0) [37,38].

## 3. Results

### 3.1. HiSeq Sequencing Output and Amplicon Sequence Variant Analysis

Regarding the bacterial 16S rRNA gene sequences, a total of 3,280,457 pairs of reads were obtained from 42 samples. Following quality control and splicing, 2,127,470 clean reads were generated from the two-terminal reads, with an average of 50,654 clean reads per sample. The sequence length was less than 450 bp (Table 1). For the fungal ITS region sequences, 3,361,862 pairs of reads were obtained from the same samples by sequencing. Following quality control and splicing, 3,050,660 clean reads were produced from the two-terminal reads, with at least 66,422 clean reads generated from each sample and an average of 72,635 clean reads. The sequence length ranged from 190 bp to 460 bp (Table 1). The results of the endophytic bacteria dilution curve showed that it tended to be stable with an increase in the number of reads sampled (Appendix A), indicating that there was a low possibility of undetected sequences and a reasonable amount of sequencing data, which could provide a number of sequences sufficient for microbial diversity analysis.

### 3.2. Effect of Jujube Witches’ Broom on Microbial Community Structure

To explore the influence of the occurrence of JWB disease on the diversity of the jujube endophyte microbial community, an analysis of α-diversity was conducted among healthy and diseased jujube samples. The results showed that the Chao1 index of endophyte bacteria in the diseased branches was significantly higher than in the healthy ones (Chao1: HB < DB, *p* < 0.05) (Figure 1a). The Shannon index of endophyte bacteria in the diseased branches and roots was significantly lower than in the healthy ones (Shannon: HB < DB, HR < DR, *p* < 0.01) (Figure 1b). The Chao1 index of endophytic fungi in the healthy roots was significantly higher than in the diseased ones (Chao1: HR > DR, *p* < 0.05) (Figure 1c). Interestingly (Figure 1d), the Shannon index of endophytic fungi in the various parts of jujube did not differ; only the Shannon index of the root part increased in the diseased samples, the species diversity increased, and the Shannon index of the other parts decreased.

To examine the impact of the occurrence of JWB disease on the jujube endophytic community structure, the Bray–Curtis distances between healthy and diseased samples were compared through principal coordinate analysis (PCoA). The quality of the PCoA results was determined by using a permutational multivariate analysis of variance (PERMANOVA). The results showed that there were significant differences in the endophytic bacterial community structure in the leaves (*p* < 0.05), branches (*p* < 0.01), and roots (*p* < 0.01) of jujube (Figure 2a). Regarding endophytic fungal community structure, only the roots exhibited a significant difference in structure (*p* < 0.05) (Figure 2b). The above results indicate that the occurrence of JWB disease changed the endophytic community structure of jujube in diverse ways in the different organs of the plant.

In addition, the α-diversity (Appendix A) and β-diversity (Appendix A) of endophytic communities in the eight groups of samples were analyzed. The different organs of jujube exhibited variable community diversity and structures. The results revealed that ecological niches can exert an influence on the community diversity and structure of endophytes.

### 3.3. Effect of Jujube Witches’ Broom on Endophytic Community Composition

A total of 44,768 amplicon sequence variants (ASVs) were detected in bacteria, and 3767 ASVs were detected in fungi. Among the endophytic bacteria groups (Figure 3a), the number of common and unique ASVs was 67. The number of total and unique ASVs in healthy leaves was higher than in diseased ones (HL > DL). However, in healthy branches, roots, and tissue-cultured shoots, the number of total and unique ASVs was lower than in those of JWB-diseased ones (HB < DB, HR < DR, TCHS < TCDS) (Figure 3a). Among the endophytic fungi groups (Figure 3b), the number of common and unique ASVs was 20. The number of total and unique ASVs in the healthy leaves was lower than the diseased ones (HL < DL); in comparison, in the healthy branches and tissue-cultured shoots, this number was higher than those in the diseased (HB > DB and TCHS > TCDS). The number of total ASVs in the healthy roots was higher than in the diseased ones (HR > DR), yet the number of unique ASVs was lower in healthy roots (HR < DR) (Figure 3b).

By utilizing high-throughput sequencing, a total of 50 endophytic bacterial phyla and 14 endophytic fungal phyla were identified. Overall, the most abundant groups of endophytic bacterial phyla in the different parts of the jujube trees were Proteobacteria (53.3–24.8%), followed by Firmicutes (46.9–16.5%), Bacteroidota (18.2–11.8%), Actinobacteriota (13.4–1.7%), and Acidobacteriota (2.4–0.8%). Regarding jujube tissue-cultured shoots, the most abundant groups of endophytic bacterial phyla were Proteobacteria, Firmicutes, and Bacteroidota (Figure 4a). In the JWB-diseased samples, the relative abundance of Proteobacteria was lower than in the healthy ones. The relative abundance of Firmicutes in the diseased samples was higher than in the healthy ones. The components of the bacterial phylum in different parts of the jujube trees varied. In healthy branches and roots, the most abundant phylum was Proteobacteria, while in diseased branches and roots, the most abundant phylum was Firmicutes. The relative abundance of Bacteroidota in the diseased roots was higher than in the healthy roots. Yet in other parts of the tree, the relative abundance was lower than in the corresponding organs of the diseased trees. The composition of endophytic fungal communities was relatively consistent among the samples at the phylum level. Unclassified fungi and Ascomycota were the phyla with the highest relative abundances, that in almost all samples exceeded 15%. Other fungal phyla such as Basidiomycota, Chytridiomycota, and Mortierellomycota were also found to have relative abundances in some samples; however, they were far lower than those of unclassified fungi and Ascomycota (Figure 4c).

At the genus level of endophytic bacteria, in the healthy leaves (HL), branches (HB), roots (HR), and tissue-cultured shoot (TCHS) samples, the dominant bacterial genera were *Escherichia* and *Shigella* (6.7%), *Escherichia* and *Shigella* (4.4%), *Rhodomicrobium* (5.4%), and *Pseudomonas* (43.7%), respectively. In addition, in each of the diseased samples, ‘*Ca*. Phytoplasma’ was the dominant bacterial genus (Figure 4b). At the genus level of endogenous fungal genera, the dominant fungal genus in various parts of the jujube was unclassified fungi. *Pichia* was the dominant fungal genus in jujube tissue-cultured shoots, and unclassified fungi were the subdominant fungal genus (Figure 4d). Jujube witches’ broom presence did not change the composition of the dominant fungal genera in each sample; however, it did change their relative abundances.

A differential analysis was conducted on each group of genera, with the result revealing the genera with differences among the groups. Regarding endogenous bacterial genera, in the leaf samples, the abundances of *Methylomonas*, *Methylophilus*, *Thiobacillus*, and unclassified *Thermodesulfovibrionia* were significantly decreased in the DL samples; in comparison, the abundance of the remaining differential genera significantly increased. In the branch and root samples, ‘*Ca*. Phytoplasma’ represented the differential genus with the highest abundance. The abundances of the remaining differential genera were all decreased. In the jujube tissue-cultured shoot group, unclassified *Lachnospiraceae* represented the differential genus with the highest abundance. The abundance of the remaining differential genera decreased in the TCDS (Figure 5a). Regarding endophytic fungi, the abundances of the differential genera in the leaf, branch, and root parts were all decreased with the occurrence of jujube witches’ broom. No obvious differential genera were noted in the tissue-cultured jujube shoot samples (Figure 5b).

In the *in vitro* culture environment, both healthy and diseased jujube shoots exhibited different endogenous bacterial and fungal diversity and composition compared with field tree organs. The number of ASVs in the *in vitro* shoots was much lower than that in the field tree organs (Figure 3); in comparison, the relative abundance of the genus *Pseudomonas* in the *in vitro* shoots was much higher (Figure 4b). The presence of JWB phytoplasmas also altered the diversity and composition of endogenous microorganisms. In the endophytic bacteria group, the number of ASVs in tissue-cultured diseased shoots was much higher than in the healthy shoots (TCDS > TCHS) (Figure 3a). At the phylum level, the relative abundance of the bacterial phylum Firmicutes in diseased *in vitro* shoots was much higher than that in the healthy shoots; in comparison, those of the phyla Proteobacteria and Bacteroidota were lower. At the genus level, despite the increase in the relative abundance of the genus ‘*Ca*. Phytoplasma’, the abundance of the genera *Pseudomonas* and *Bacteroides* decreased (Figure 4b).

### 3.4. The Impact of Jujube Witches’ Broom on the Functions of the Endophytic Microbial Community

The functional identification of the endophytic bacterial community in jujube was carried out using the PICRUSt2 software. In JWB-diseased jujube, the relative abundances of bacteria with the functions of replication, translation, and carbohydrate metabolism in branches and roots were higher than in the diseased parts (DB > HB and DR > HR). In comparison, the relative abundance of bacteria with the functions of signal transduction, metabolism of cofactors and vitamins, and amino acid metabolism in the healthy branches and roots was higher (DB < HB and DR < HR). In the leaf samples and jujube tissue-cultured shoot samples, the bacterial composition with each function did not significantly differ. The abundance of bacteria with the functions of signal transduction, metabolism of cofactors and vitamins, and membrane transport was relatively higher in the jujube tissue-cultured shoot samples; in comparison, the relative abundance of global and overview maps was the highest in the leaf samples (Figure 6a).

The functional prediction of the fungal community was performed using the FUNGuild software (Version 1.0). The results indicate that fungi with unassigned functions have the highest relative abundance in the three parts of the jujube trees. In the diseased leaf (DL) samples, the relative abundance of fungal function in Animal Pathogen-Plant Pathogen-Undefined Saprotroph was 1.7% higher than in healthy leaves (HL); in comparison, the relative abundances of Dung Saprotroph-Undefined Saprotroph-Wood Saprotroph and Endophyte-Lichen Parasite-Plant Pathogen-Undefined Saprotroph were 4.5% and 3.3% lower, respectively. In the diseased branch (DB) samples, the relative abundance of fungi with the function of Animal Pathogen-Plant Pathogen-Undefined Saprotroph was 8.1% higher than in the healthy branches (HB), and the relative abundances of fungi with the function of Endophyte-Lichen Parasite-Plant Pathogen-Undefined Saprotroph and Animal Pathogen-Endophyte-Epiphyte-Plant Pathogen-Undefined Saprotroph were 1.8% and 2.2% lower, respectively. In the diseased root samples, the relative abundance of fungi with the function of Animal Pathogen was 4.5% less than in the healthy roots (HR), and fungi with the function of Plant Pathogen-Undefined Saprotroph (3.2%) emerged in the DR samples. In the tissue-cultured diseased shoots (TCDS), fungi with the function of Animal Endosymbiont-Animal Pathogen-Plant Pathogen-Undefined Saprotroph were dominant (>50%), and their relative abundance was 4.5% less than that in healthy ones (TCHS). The above results indicate that jujube witches’ broom alters the functions of the endophytic microbial community in jujube (Figure 6b).

### 3.5. The Impact of Jujube Witches’ Broom on the Co-Network of Endophytic Bacteria–Fungi in Jujube

The genera with a relative abundance higher than 1% were selected for analysis and simultaneous modular division on the correlation network graph. In the healthy group, the network graph contained 209 nodes, 1201 edges, and 10 modules. Module 1 was the largest module, containing 64 genera, all of which were bacterial genera (Figure 7a). In the diseased group, the network graph contained 185 nodes, 1767 edges, and 7 modules. Module 1 was the largest module, containing 40 bacterial genera and 11 fungal genera. ‘*Ca*. Phytoplasma’ was present in this module and exhibited a negative correlation with *Bradyrhizobium*, *Clostridium sensu stricto 1*, *Subdoligranulum*, *unclassified Caulobacteraceae*, *unclassified Clostridia*, and *unclassified Lachnospiraceae*. Following the occurrence of jujube witches’ broom, compared with the healthy group, the interaction network graph of the diseased group showed a decreased number of nodes and modules but an increased number of edges. This may indicate that jujube witches’ broom induces structural and functional remodeling in the interaction network, shifting from a relatively complex state of multiple modules, multiple nodes, and sparse connections to a state of simplified modules, streamlined nodes, and tight connections.

## 4. Discussion

A number of research groups have reported that plant endophytic microbial community diversity and structure are altered by plant pathogens, such as the Chinese wheat mosaic virus [39], *Bursaphelenchus xylophilus* [40], and *Exobasidium vexans* Masee [41]. Studies on the microbiome community of diseased plants of Chinese chestnut with yellow crinkle disease associated with ‘*Ca*. P. castaneae’ [9], the apple proliferation disease associated with ‘*Ca*. P. mali’ [13], and grapevine infected with “flavescence doré” phytoplasma [42] have revealed that infection with phytoplasmas alters endophytic microbiome diversity and composition. In this study, through high-throughput sequencing, it was found that the diversity, structure, and function of the endophytic microbiome community in both field jujube trees and *in vitro* shoots are diverse in the presence of phytoplasmas. Significant differences in the bacterial community component were found in the branch and root samples, possibly related to the presence of phytoplasmas.

In chestnut infected by phytoplasmas, the bacteria Alphaproteobacteria, Sphingobacteriaceae, *Pseudomonas*, and *Stenotrophomonas* and the fungi Pleosporaceae, *Alternaria*, and *Eurotiomycetes* showed significant differences among diseased and healthy samples [9]. In grapevine, the bacteria *Burkholderia*, *Methylobacterium*, and *Pantoea* were affected by the presence of phytoplasma infection [13]. In this study, the relative abundance of bacteria of the phyla Proteobacteria, Firmicutes, and Bacteroidota was modified by the presence of JWB phytoplasmas. The genus ‘*Ca*. Phytoplasma’ became dominant with the highest abundance in JWB-diseased samples. The effect of phytoplasma infection on the fungal community seems less relevant. The bacterial phyla detected in plants infected by phytoplasmas are similar to those reported in citrus trees infected by “huanglongbing” (‘*Ca*. Liberibacter asiaticus’) [43].

The occurrence of JWB disease also impacts the biochemical and physiological processes in jujube plants, such as phytosynthesis activity, mineral elements, and phytohormone contents [14]. Despite the direct influence of JWB phytoplasma on jujube endophytes, these biochemical and physiological changes may also have an indirect influence on endophyte diversity and components.

For reasons related to cost, in the current study, it was only possible to perform sampling once in the orchard trees and *in vitro* shoots for endophyte analysis. However, by employing high-throughput sequencing techniques, large amounts of data were obtained. The number of ASVs was much higher than the number of endophyte microbial species identified using traditional isolation methods [25]; moreover, the sequencing data also provide a basis for further analysis in jujube and endophyte interaction studies.

In general, endophytes can have neutral or detrimental effects on the host plant under normal growth conditions; in comparison, they can be beneficial under more extreme conditions or during different stages of plant life [44]. As pathogenic bacteria, phytoplasmas inhabit the phloem of host plants and are transmitted by sap-feeding insect vectors. Symptoms induced by phytoplasmas, such as witches’ broom and phyllody, increase the prevalence of short branches and small young leaves, enhancing the attraction of insect vectors and thus the spread of phytoplasmas. Such manipulations of the morphology of host plants appear to be a common strategy for the survival of phytoplasmas [45]. In the case of jujube witches’ broom, the presence of phytoplasmas not only manipulates the development of jujube plants, but also affects the diversity and structure of endogenous bacterial and fungal species, which also leads to a change in the function of endophytes.

Interactions between microorganisms and plants may trigger an immune response, inducing plant defense reactions against pathogens. The application of antagonistic microorganisms has resulted in remarkable success in plant protection practice. Through the screening of antagonistic bacteria, *Brevibacterium halotolerans* JZ7 was developed into a biological control agent to combat root rot disease in jujube [25]. In the current study, the relative abundance of fungi with a function of plant–pathogen interaction was found to increase in the diseased samples. The interaction network of the endophytic microbiome community also changed. Due to the difficulties encountered in the pure culture of JWB phytoplasma, it is still not possible to perform the dual culture of phytoplasma with other endophyte microbiomes. In future studies, it will be necessary to explore in depth the interactive relationships between endophytes and phytoplasmas, screen out combinations of endophytes that have an efficient antagonistic effect against phytoplasmas and are able to promote the plant growth, and identify microorganisms for JWB disease biocontrol.

Owing to the difficulties in phytoplasma cultivation in artificial media, the *in vitro* culture of diseased shoots is employed for phytoplasma maintenance in collections [46]. During the establishment of *in vitro* cultures, the plant tissue is surface sterilized. Endophytic bacteria and fungi living inside the tissue survive this process and persist in the material [47]. In a recent study, the cultivation of the JWB phytoplasma ‘*Ca*. P. ziziphi’ in both liquid and solid media from *in vitro* JWB-diseased shoots was achieved together with *Pseudomonas* species [48], a result that is in agreement with the finding in the same tissues of this bacterium in both healthy and JWB-diseased in vitro shoots. The function of *Pseudomonas* and its interaction with ‘*Ca*. P. ziziphi’ requires further investigation to obtain pure colonies of each bacterium and elucidate the possible functional interaction together with the pathogenetic mechanisms of JWB phytoplasmas.

## 5. Conclusions

In this study, infection with JWB phytoplasmas was found to modify the abundance, diversity, and community structure of endophytic bacteria and fungi in jujube. In the branches and roots, the richness of endophytic communities significantly increased, and the diversity decreased in the presence of JWB phytoplasmas. The phyla Proteobacteria, Firmicutes, and Bacteroidota and the genus ‘Ca. Phytoplasma’ were the most abundant phyla and genera in the diseased branches and roots. The presence of JWB phytoplasma is also associated with the remodeling of the endophytic microorganisms’ interaction network, that resulted in a shifting to a simpler state. This analysis revealed changes in the diversity, composition, and function of the endophyte microbiome community in jujube in the presence of JWB phytoplasmas. The results also shed light on the possibility of developing biological control agents and obtaining pure cultures of JWB phytoplasmas.

## Figures and Tables

**Figure 1 microorganisms-13-01371-f001:**
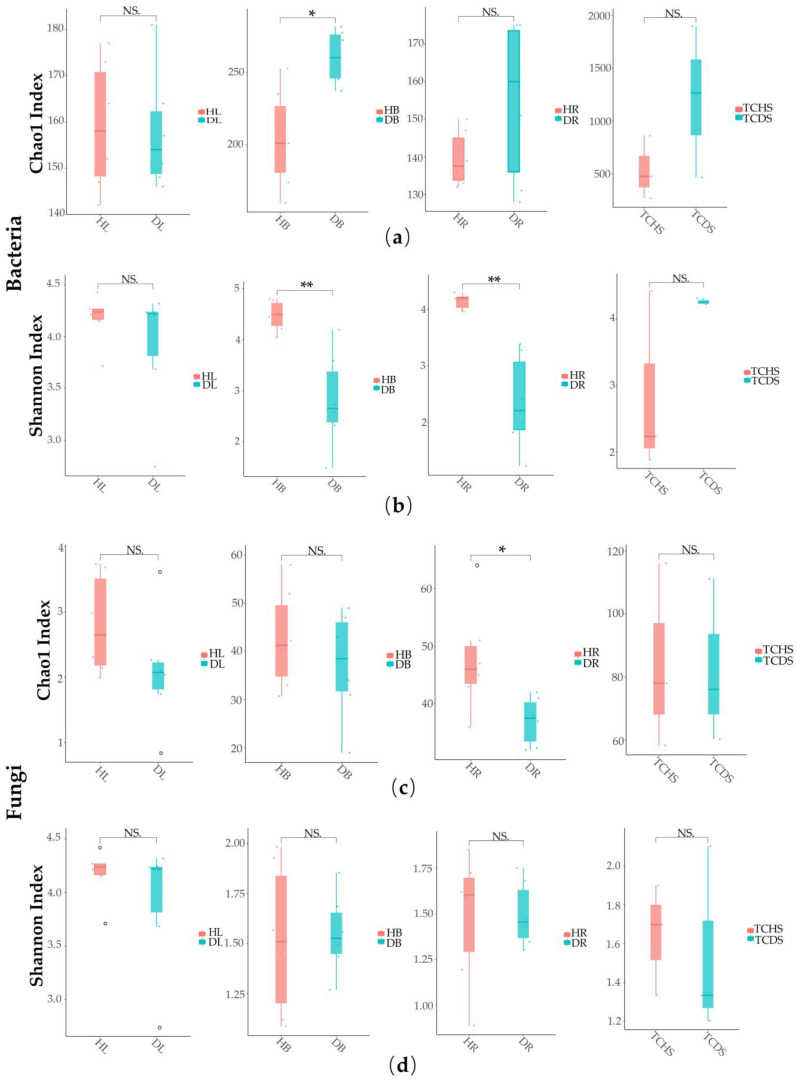
Analysis of the α-diversity of endophytic bacteria and fungi in jujube. (**a**,**c**) Chao1 index and (**b**,**d**) Shannon Index. The Chao1 and Shannon indices of each sample repeat are presented in dots of certain colors. Outlier data are presented in circles. * indicates a significant difference at the *p* < 0.05 level. ** indicates a significant difference at the *p* < 0.01 level. NS indicates no difference at the *p* ≤ 0.05 level.

**Figure 2 microorganisms-13-01371-f002:**
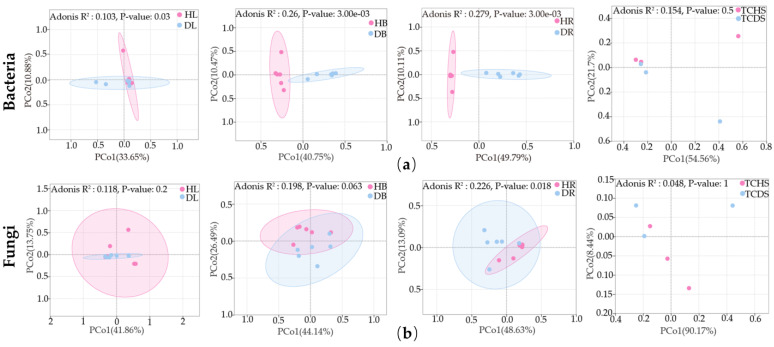
Principal component analysis of the endophytic microbial community in jujube. (**a**) Endophytic bacteria and (**b**) endophytic fungi. Each point in the figure represents a sample; different colors and shapes represent different samples/groupings; the oval circles indicate that they are 95% confidence ellipses; and the confidence ellipses show when the number of samples is higher than three. The abscissa represents the first principal component, and the percentage indicates the contribution value of the first principal component to the sample differences. The ordinate represents the second principal component, and the percentage indicates the contribution value of the second principal component to the sample differences.

**Figure 3 microorganisms-13-01371-f003:**
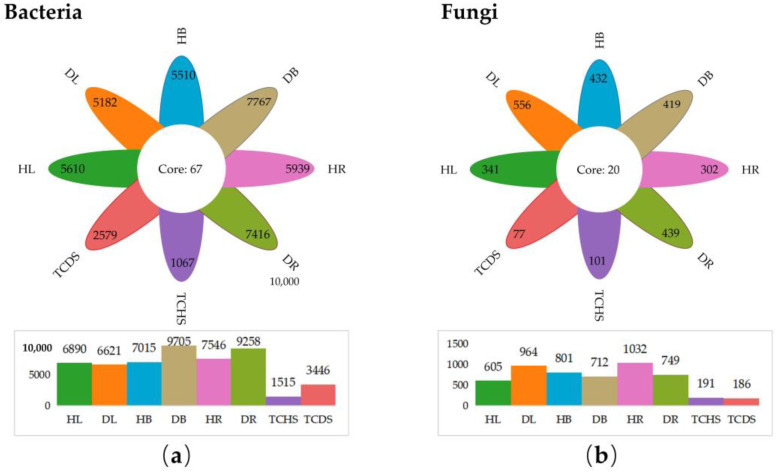
The Venn diagram merges with the columnar statistical chart, showing the unique, shared and total (**a**) bacterial ASVs and (**b**) fungal ASVs among the different samples.

**Figure 4 microorganisms-13-01371-f004:**
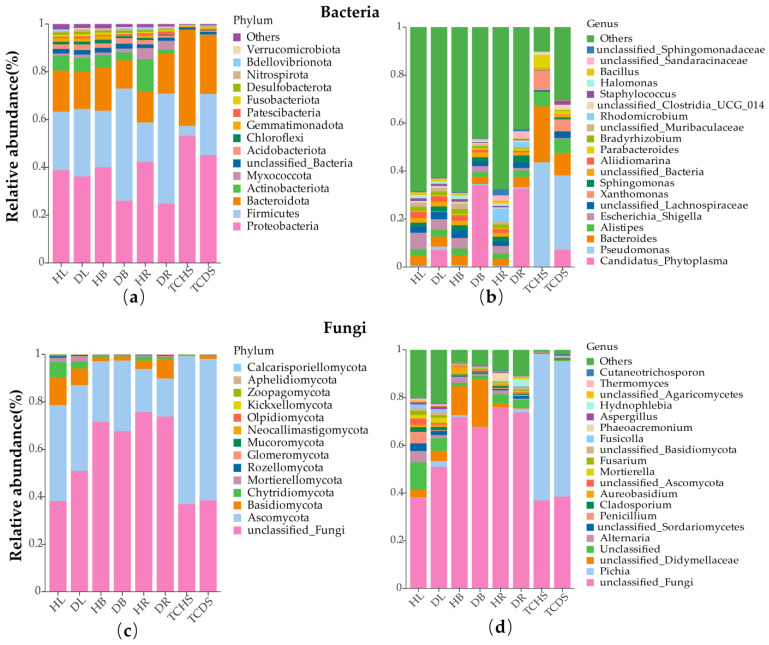
The impact of jujube witches’ broom on the composition of the endophytic microbial community at (**a**) the phylum level of endophytic bacteria; (**b**) the genus level of endophytic bacteria; (**c**) the phylum level of endophytic fungi; and (**d**) the genus level of endophytic fungi.

**Figure 5 microorganisms-13-01371-f005:**
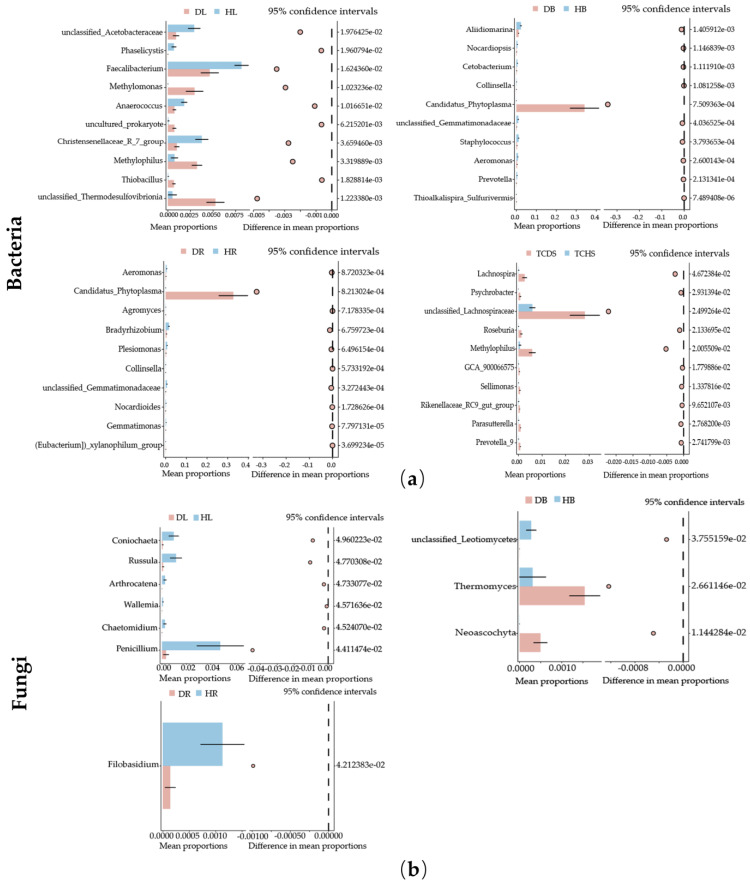
Analysis of the differences in endophytic microbial genera levels between the groups: (**a**) endophytic bacteria and (**b**) endophytic fungi. Different colors in the figure represent different groups or samples. In each picture, the graph on the left-hand side illustrates the abundance proportions of different species in two samples or two groups of samples. The middle section illustrates the different proportions of species abundances within the 95% confidence interval. The value on the far right is the *p*-value.

**Figure 6 microorganisms-13-01371-f006:**
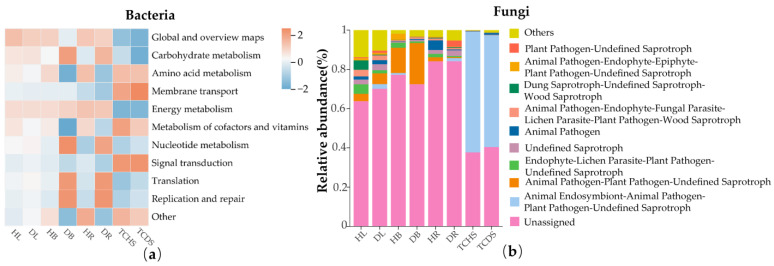
The impact of jujube witches’ broom on the functions of the endophytic microbial community in jujube: (**a**) endophytic bacteria and (**b**) endophytic fungi.

**Figure 7 microorganisms-13-01371-f007:**
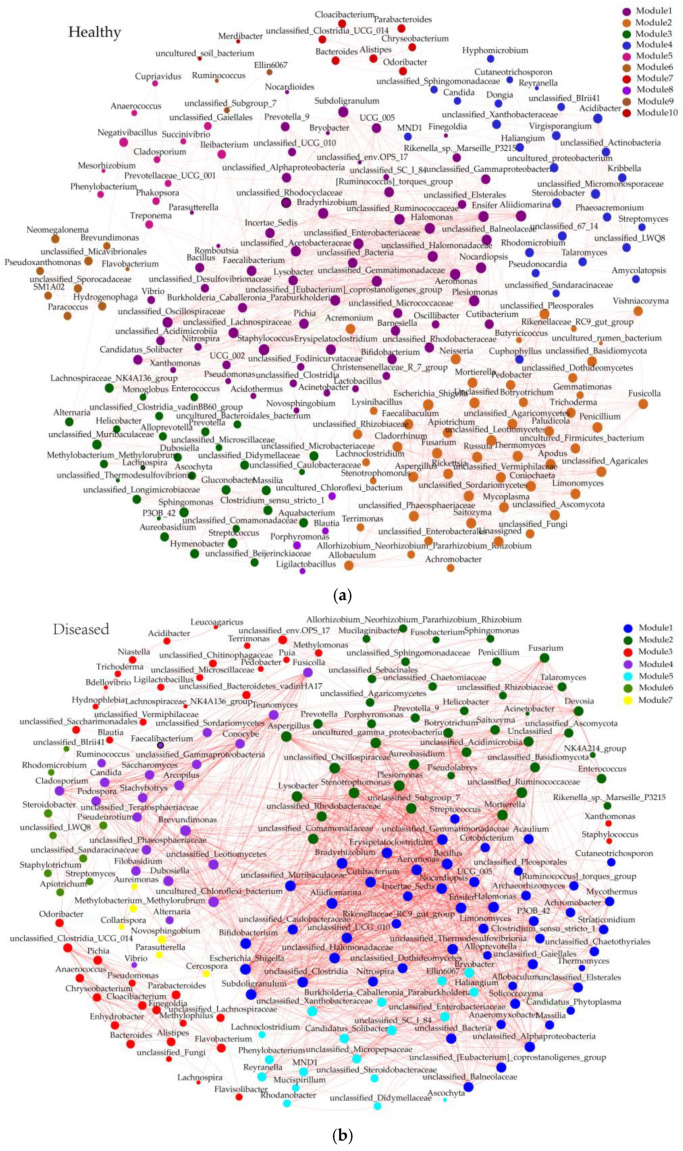
Correlation network analysis of endophytic bacterial–fungal communities in jujube at the genus level. (**a**) Analysis of the interactions between endophytic bacteria and fungi in the healthy group and (**b**) analysis of the interactions between endophytic bacteria and fungi in the diseased group.

**Table 1 microorganisms-13-01371-t001:** Statistics of jujube endophyte sequencing data processing results.

Name	Raw Reads	Clean Reads	Sequence Length	Amplicon Sequence Variants (ASVs)
Bacteria	3,280,457	2,127,470	Less than 450 bp	44,768
Fungi	3,361,862	3,050,660	190 bp to 460 bp	3767

## Data Availability

The original contributions presented in this study are included in the article/Appendix A. Further inquiries can be directed to the corresponding authors.

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
