# Peer review of "The Impact of Jujube Witches’ Broom Phytoplasma on the Community Structure of Endophytes in Jujube"

_microorganisms, 2025, doi:10.3390/microorganisms13061371_

Round 1

Reviewer 1 Report

Comments and Suggestions for Authors

This is an interesting manuscript and the study is largely well conducted. The logic of the research is a little confusing, however. It is worthwhile studying the impact of phytoplasma on microbial diversity in jojoba but is this contributing to the control of plant pathogens or simply exploring aspects of microbial ecology? Both are worthwhile motivations for the study but this should be clearly articulated.  The English expression, grammar and spelling throughout needs considerable attention.

Comments on the Quality of English Language

Needs some attention.

Author Response

Thank you very much for taking time to review this manuscript. We appreciate your comments on the manuscript.

The aim of this study was to evaluate the influence of JWB phytoplasma on the composition, diversity, and potential function of the jujube endophytic community. Our study results could be used to evaluate potential biological control agents for JWB disease. We also inferred the interaction relationship between the endophytic microbiome and JWB phytoplasma. These information were added to the end of the introduction part. See Line 92-97 in the clean pdf version.

The manuscript went through careful revision to correct the grammatical mistakes. We also employ the English Editing Service from MDPI for better expression.

We believe the manuscript is now improved and thank you for the suggestions and help in its revision.

Reviewer 2 Report

Comments and Suggestions for Authors

The topic of the manuscript falls into the scope of the Microorganisms MDPI Journal. The studies of endophytes has become more and more common these days proving the importance of the topic. Each new plant-microbial relationship described is important, fulfilling the knowledge as well as providing new microbes beneficial in food production and other applications.

In this manuscript the perspective is reversed - an effect of pathogenic JWB phytoplasma on microbial structure is demonstrated what is of high novelty to me as most of works focus on beneficial microbes.

The whole manuscript is well prepared and have good quality. All information in methods are provided. Results are clearly described and supported by good figures. All combined with good discussion leads to convicing results.

Detailed comments:

  • I cannot find Supplementary materials to see data and results
  • Line 107: please add space in 'software[27]'
  • Line 108: please add space and removed not necessary in '(Version 1.9.1 )[28]'
  • Line 113: please correct SILAV to SILVA
  • Line 117: in the sentence 'The diversity and β diversity...' you probably forgot to mention alpha diversity - 'The α and β diversity...'
  • Line 126: please add space in 'GGPLOT 2R package'
  • In methods I miss information what tools were used for Venn diagrams generation and for functional analysis (FAPROTAX and FUNGuild) as well as of Correlation Network. Additionally, you should mention the usage of such tolls and analysis in in methods not in results. By the way in text the word 'FUNGuiild' is used instead of FUNGuild.
  • The Fig. 1 caption should be placed below the figure 1
  • The caption of Fig. 1 suggest that * and ** present significant correlations between diversity indices. However, Kruskal-Wallis H test was used. I think this figure presents differences between samples not correlations. Please check this.
  • Line 160: I'd use past tense here, as this was done prior writing the text: "To investigate the impact of the occurrence of JWB disease on the jujube endophytic community structure, the Bray-Curtis distances between healthy and diseased samples are were compared by Principal Coordinates Analysis (PCoA)."
  • Line 195: a period after 'and total.' is not required
  • Line 231-232, 236: please italize the bacteria genera
  • Figure 5 has very small font making it hard to read
  • You use both 'in vitro' and 'in-vitro'. Please unify this

Reviewer 3 Report

Comments and Suggestions for Authors

The manuscript entitled “The impact of jujube witches' broom on the community structure of endophytes in jujube” has been reviewed in detail. This study investigates how jujube witches' broom (JWB) phytoplasma affects endophytic microbial communities in jujube trees. The researchers employed high-throughput sequencing of bacterial 16S rRNA and fungal ITS regions to characterize endophytic communities in healthy and JWB-infected field trees and in vitro shoots. They analyzed diversity metrics, community composition, predicted functional capabilities, and microbial interaction networks between healthy and diseased samples. Results revealed that JWB infection significantly altered endophytic community structure, particularly in branches and roots. These findings demonstrate how pathogen invasion can restructure complex microbial networks within plants.

  • For a general reader, please write the complete term before using its abbreviation. Afterwards, write the abbreviation of that term concerned. Use this practice throughout the manuscript.
  • Please go through the whole manuscript and correct grammatical mistakes
  • Please improve the abstract
  • Please write keywords based on alphabetical order
  • The manuscript references using "SILAV and UNIT as a reference database," which appear to be typographical errors for the standard SILVA and UNITE databases. Correct the database names to SILVA and UNITE, and specify the exact database versions used.
  • Describe how taxonomic conflicts or ambiguities were resolved, particularly for closely related taxa
  • Replace phrases like "JWB increased" or "JWB decreased" with "JWB was associated with increased" or "JWB-infected plants showed decreased"
  • Discuss alternative explanations for observed microbiome differences, such as: Plant physiological responses to infection that may indirectly alter microbiome composition
  • Provide complete primer sequences in the main text or make the supplementary Table readily available
  • Include a map of the orchard showing the spatial distribution of sampled trees, clearly indicating the position of healthy and diseased specimens
  • Document key environmental variables for each sampled tree (e.g., soil properties, microclimate conditions, irrigation access)
  • Provide a detailed justification for why 6 trees per condition was selected, referencing studies with similar designs if applicable
  • Implement validation tests for the surface sterilization protocol
  • Figure 4: it is written” (a) Phyla level of endophytic bacteria; (b) Phyla level of endophytic fungus. (c) Genus level 227 of endophytic bacteria; (d) Genus level of endophytic fungus. “but (a) and (b) depict bacteria, (c) and (d) depict fungi
  • Please improve the discussion
Comments on the Quality of English Language
  • Please go through the whole manuscript and correct grammatical and typo mistakes

Reviewer 4 Report

Comments and Suggestions for Authors

The manuscript microorganisms-3636890, titled “The impact of jujube witches' broom on the community structure of endophytes in jujube” addresses an important paper investigating how jujube witches’ broom (JWB) phytoplasma infection affects the abundance, diversity, community structure, and interaction networks of endophytic bacteria and fungi in both field-grown and in vitro jujube trees..  However, in my opinion this paper must be revised in a major manner for reasons of forms and content.

Introduction

Could the authors provide more recent references (from the last 3–5 years) to better contextualize the current state of research?

How does this study build upon or differ from previous studies? Clarifying this could strengthen the novelty of the work.

Could the authors state the hypothesis or specific research questions more explicitly at the end of the introduction?

Materials and Methods

Are the microbial strains used in the study deposited in a publicly accessible culture collection? If so, please provide accession numbers.

How many biological and technical replicates were used per treatment, and how were they randomized?

Results

Some results are described without clear reference to corresponding figures or tables. Could the authors improve the linkage between text and visuals?

Were any outliers observed in the data? If so, how were they handled?

Discussion

Could the authors better integrate their results with findings from other studies? A comparative analysis could enhance the manuscript.

Are there limitations to the study (e.g., sample size, environmental conditions, strain specificity) that should be acknowledged?

What are the implications of this research for future field applications or commercial scaling?

Round 2

Reviewer 1 Report

Comments and Suggestions for Authors

There are still some issues of English expression throughout the manuscript.

Comments on the Quality of English Language

There are still some issues of English expression throughout the manuscript.